# Surveillance Program of Clinical Samples for Polio and Non-Polio Enteroviruses in the Slovak Republic during the 1958–2020 Period

**DOI:** 10.3390/v14091957

**Published:** 2022-09-03

**Authors:** Renata Kissova, Katarina Pastuchova, Viera Lengyelova, Marek Svitok, Jan Mikas, Shubhada Bopegamage, Cyril Klement

**Affiliations:** 1Department of Medical Microbiology, Regional Public Health Authority Banska Bystrica, Cesta k Nemocnici 25, 97401 Banska Bystrica, Slovakia; 2National Reference Laboratory of Poliomyelitis Public Health Authority, Trnavska Cesta, 82102 Bratislava, Slovakia; 3Department of Medical Microbiology, Regional Public Health Authority Senny Trh, 82009 Kosice, Slovakia; 4Faculty of Ecology and Environmental Sciences, Technical University in Zvolen, T. G. Masaryka 24, 96001 Zvolen, Slovakia; 5Public Health Authority, Trnavska Cesta, 82102 Bratislava, Slovakia; 6Enterovirus Laboratory, Institute of Microbiology, Faculty of Medicine, Slovak Medical University, Limbova 12, 83303 Bratislava, Slovakia; 7Faculty of Public Health, Slovak Medical University, Limbova 12, 83303 Bratislava, Slovakia

**Keywords:** clinical samples, surveillance, polioviruses, non-polio enteroviruses

## Abstract

Enteroviruses (EVs) are associated with a wide spectrum of diseases involving various organs. Our aim was to give a historical overview of the genesis of clinical sample processing for EVs in the Slovak Republic (SR) during the 1958–2020 period, within the framework of the World Health Organization (WHO) polio program. Further, analyses were made of the data obtained from the archives of processed clinical sample surveillance using statistical methods. We used generalized additive models (GAM) with binomial distribution and logit link functions and an autoregressive moving average (ARMA) to analyze the data obtained during this 63-year period. Our results show trends in the composition of EV strains circulating in the population. Furthermore, statistically significant increasing trends of the non-polio enteroviruses (NPEVs) were observed over the studied time, represented by echoviruses (E) and coxsackieviruses A and B (CVA and CVB), with a cyclical pattern of occurrence. The most prevalent serotype over this period was CVB5, which became significantly more prevalent after 2000. While PVs, CVB1, and CVB3 were present in the second half of the studied period, CVA10, CVA16, E3, E25, and E30 appeared more frequently.

## 1. Introduction

Enteroviruses (EVs) are transmitted mainly by the fecal–oral and respiratory routes. They are excreted in the stool over a period of days to weeks after the acute phase of infection [1]. These viruses are known to contaminate mainly waste and surface waters [2]. Personal hygiene and drinking water from safe sources are important preventive measures [3,4]. EV infections are accompanied by a wide spectrum of clinical symptoms, which can range from mild to life-threatening. Diseases caused by EVs can be divided into neurological diseases such as aseptic meningitis, neuritis, acute flaccid paralysis, and encephalitis; generalized diseases with fever, involvement of mucous membranes, and skin; inflammatory diseases of the nasopharynx and upper respiratory tract or conjunctiva; and diseases of internal organs such as myocarditis, pancreatitis, and severe neonatal sepsis [3,4]. EVs are also associated with chronic diseases such as type 1 diabetes [4,5]. PV infections may clinically be inapparent and mild; on the other hand, they may appear as aseptic meningitis and develop into paralytic poliomyelitis. More serious clinical manifestations of PVs can affect the brain and spinal cord. The most serious form is poliomyelitis, which may result in lifelong disabilities and death. PV1 has greatest paralytic potential [6], whereas, non-polio enteroviruses (NPEVs) such as EV68-71, echoviruses (E), coxsackieviruses A (CVA), and coxsackieviruses B (CVB) serotypes can cause diseases with clinical signs and manifestations similar to those caused by other EV serotypes. Paralytic diseases and acute flaccid paralysis (AFP) can also occur as a consequence of NPEV infection. In addition, EV71, EV70, some Es, and CVs such as CVA7 are strongly neurotropic [4]. In terms of public healthcare, the problem of poliomyelitis has undeniably been at the forefront in the past, making the PVs most significant EVs, especially in 1950s.

In the Slovak Republic (SR), compulsory reporting of polio was introduced as early as 1928 [7]. The World Health Organization (WHO) polio program was initiated in 1953 [8]. European regional laboratories underwent training for cell culture work in 1957–1958 [8]. In 1985, the Global Polio Eradication Initiative (GPEI) program was established by the WHO. On 12 May 1988 at the 41st session of the World Health Assembly (WHA), the WHO announced a program to eradicate poliomyelitis [9]. Monitoring of acute flaccid paralysis (AFP) in children under 15 years of age, with complementary environmental surveillance of PVs, is a part of the poliomyelitis eradication program. This program allows surveillance of not only the PVs but also the NPEVs. The SR has been involved in this program since its establishment [2,10,11,12]. Initially, it was associated with the monitoring of paralytic diseases caused by PVs and the monitoring of the effectiveness of vaccination against poliomyelitis. Introduction of the compulsory vaccination of children against poliomyelitis by the Salk vaccine (inactivated polio vaccine; IPV) occurred in Czechoslovakia in 1957 [13], and later, by the 1960s, the Sabin (live attenuated) oral polio vaccine (OPV) [14], as shown in Appendix A. The SR and the Czech Republic (previously Czechoslovakia/ Czechoslovak Socialist Republic (CSR)), were among the first countries in the world to eliminate the paralytic form of polio, due to a widespread vaccination program [15]. In the year 1960, monitoring and reporting of the so-called polio mimic diseases, today known as acute flaccid paralysis (AFP), also began [14]. The last paralytic poliomyelitis case in the former CSR was seen in the year 1960 [14].

Vaccination with the live OPV was carried out on a campaign basis in the spring months in children from the 10th week of life over two consecutive years, always in two stages during March and May. Vaccination of school-age children has since been carried out annually as part of the second round of vaccinations in May (Appendix A). Until 1992, the effectiveness of vaccination was monitored annually by determining immunity against poliomyelitis in about 1500 people. The SR maintained a high vaccination rate (97%) [16]. In 2020, vaccination rates for children 2–14 years of age ranged from 95.6% to 97.4%, and in 2021 they ranged from 95.4 to 97% [17,18]. In addition, antibodies to EVs have previously been monitored in selected populations, e.g., children or workers in wastewater treatment plants [19,20].

In 2005, a change in the vaccination strategy occurred in the SR when Sabin’s live oral OPV vaccine was replaced by an inactivated IPV Salk’s vaccine. The reason was isolation of vaccine-derived PVs (VDPV type 2) in the years 2003–2005 in wastewater in the western territory of the SR (area Bratislava–Vrakuna, Skalica) [21]. Similar changes had already spread through the European countries in the Euro region since June 2002 after the Euro Region was declared Polio free. The NPEVs in the European Union (EU) and European Economic Area (EEA) region were discussed in detail by Buba et al. [22], where it was also stressed that there are differences in the surveillance systems in different countries of the EU. In some countries, NPEVs are detected as part of clinical diagnoses or in research projects and not as part of a surveillance program for NPEVs. In the SR, detection of EVs was previously entirely dependent on the AFP and sewage surveillance of the WHO and this was the only means for detecting NPEVs. Therefore, this study has its limitations in terms of NPEV surveillance as the focus of the WHO is to detect PVs. We recently reported [23] the wastewater assessment of the PVs and NPEVs during the 1963–2019 period [23].

The aim of our work was to provide a historical overview of EV diagnostics in the Slovak Republic during the years 1958–2020 using the available archival data, and also to analyze the trends in the composition of EV strains circulating in the population. In addition, we aimed to compare the obtained data with the prevalence of circulating EV strains in treated wastewater (in the SR) using recently reported data [23] and statistical methods.

## 2. Materials and Methods

### 2.1. Development of the Methodology

Testing of clinical specimens began at the National Reference Center (NRC) for poliomyelitis in Bratislava (located in the western region of the SR) in 1958 (from the earliest traced records), followed by the laboratory in Banska Bystrica (central region of the SR) in 1962 and Kosice (eastern region of the SR) in 1964 [15]. Clinical samples, which were collected from symptomatic patients, covered the whole SR. Clinical samples and wastewater were examined simultaneously for EVs from the 1960s [15,23].

### 2.2. Sample Processing

For the isolation of EVs and especially the PVs, a reference was made in the archives to the 1957 resolution of a WHO expert committee. The 1957 WHO Geneva Commission of Experts [8] declared that virus isolation is the most valid diagnostic method in the diagnosis of PVs and NPEVs. However, the material had to be collected and transported to the laboratory in a timely manner.

Stool was used as the most suitable material for isolation. The method for collection of multiple stool samples for examination (at least two samples) taken within 24–48 h was already established early in the study period. Other clinical samples investigated were cerebrospinal fluid (CSF), throat, nasal and conjunctival swabs, and autopsy materials. Specimens were collected in sterile containers, and refrigerated and transported as soon as possible after collection.

For stool processing, each sample was diluted in Earle’s medium (1:10). Samples were centrifuged and antibiotics were added to the supernatant after centrifugation. Both the original sample and the supernatant (suspension) were stored at −20 °C.

### 2.3. Virus Isolations

Cell cultures used were also based on the recommendations of the 1957 WHO expert committee [8]. In the 1960s, buffalo green monkey kidney cells (BGM) from the Institute of Serum and Vaccines, Prague, the Czech Republic (CZ), were used (three blind passages of the samples were performed before declaring a negative result). Other cell lines used were as follows: HeLa during the period of 1962–1969 (four blind passages, negative results were declared if cytopathic effect was absent throughout) and human amniotic cells (primary cultures from placenta) and human embryonic cells (institutional forms were filled out as no special ethics permissions were required in the 1960s and early 1970s). Since 1968, human embryonic lung cells (LEP19) from the Institute of Serum and Vaccines, Prague, the CZ, have been used. In 1984, Vero (African green monkey kidney) cells were added to the list for the virus neutralization test (VNT) for both CVs and PVs. Details are provided in Appendix A. During the period of 1962–1990, experiments were also carried out on suckling mice aged 12–36 h (for investigation of neuroinfections: paresis, meningitis, encephalitis). The typing of isolates was carried out at the Institute of Epidemiology and Microbiology (IEM) Prague, CZ [15]. At least two types of cell cultures were chosen to isolate the EVs. The thermoresistance, ether-resistance tests, and, since 1970, the chloroform-resistance test have been used to distinguish EV isolates from other viruses [15,24,25].

Cells were seeded at a concentration of 150–250,000 cells/mL in the growth medium (MEM- Earles) with 10% fetal calf serum (FCS), with 100 U/mL penicillin (PNC) and 100 µg/mL streptomycin (STM) [26]. In earlier years, lactalbumin hydrolysate or Medium 199 was added for cell growth promotion. Clinical material was inoculated on 24–48 h cell cultures. Three to five tubes were used for one stool specimen, and later two tubes were used.

Measures of 0.2 mL of stool suspensions (treated with chloroform, with 100 U/mL PNC and 100 µg/mL STM) [26] or undiluted CSF or other clinical material (throat, nasal, or conjunctival swabs and autopsy samples) were used for the infection of monolayers.

After 2 h adsorption, the inoculum was replaced with maintenance medium (MEM-Earles) containing 2% FCS, 100 U/mL PNC, and 100 µg/mL STM [26]. The infected cells (in cell culture tubes or flasks) were incubated for 7 days at 36 °C. Monolayers were checked for cytopathic effect (CPE) (from 24–36 h, usually till 6 days) daily. Cytotoxicity was usually evident within 24 h on blind passages. Samples which showed CPE were frozen at −20 °C before identification. Throughout the studied time period, we followed the criteria as defined by the WHO manuals (Appendix A).

Hyperimmune sera or pool sera were used to identify EVs using the virus-neutralization test (VNT). Currently, identification of PVs and NPEVs in the SR is performed via VNT using type-specific antisera (before 2015, RIVM typing pool; later, Lim and Benyesh-Melnick pool) and subsequently PCR in accordance with WHO recommendations [10,11,26,27]. All isolates with suspected polioviruses must be identified in the intratype differentiation (ITD) [28]. ITD and sequencing analysis were performed at the WHO Regional Reference Laboratory of the National Institute for Health and Welfare, Helsinki, Finland.

In cases where a long period had passed since the onset of the disease, serological techniques were used, namely serum-neutralization test and metabolic-inhibition test [15]. In 1968–1971, the complement-fixation test (CF) was also used, with the two-dimensional method of Black and Melnick [29]. However, when using these tests in laboratory diagnosis, cross-reactions between serotypes and nonspecific antibody elevations in different EVs against each other (also with polio) were very common. Therefore, serological reactions were used only as complementary testing.

### 2.4. Data Analysis

Inclusion and exclusion criteria: All available data on EV isolations from clinical samples examined in three public health laboratories over the period of 1958–2020 (63 years), covering the whole SR territory, were included in this study. This represents all clinical cases seeking treatment from the health care system, which were investigated and reported by the enterovirus surveillance system. All the archival—recorded, isolated, and subtyped—EV strains in the Slovak Republic were included in the statistical processing.

To explore long-term temporal trends in the prevalence of EVs, we used generalized additive models (GAM) with binomial distribution and logit link functions [30]. We fitted separate GAMs for four virus groups (CVA, CVB, E, and PV) and three major strains of NPEVs (CVB5, E6, and E30) using a thin-plate regression spline smoother of time to allow potential nonlinear trends in the distribution. Autocorrelation function estimates were explored to check for serial independence in the residuals of each model. In cases of significant temporal autocorrelation, the data were refitted by GAMs involving autoregressive-moving average (ARMA) correlation structures. A sequence of ARMA GAMs was fitted to each response and the model of the lowest ARMA order with a nonsignificant autocorrelation function was selected as a final model. Wald tests were used to assess the significance of the GAMs [31].

In addition, we used canonical correspondence analysis (CCA) [32] to provide an overview of changes in the composition of EV records related to time. CCA is an ordination method used to reduce a multidimensional space consisting of many species (or virus strains) by reducing the number of dimensions into a few major gradients while accounting for independent variables such as temporal trends. The gradients represented by ordination axes are the basis for visualization of changes in an assemblage composition via an ordination plot. The ordination plot is a graph with a coordinate system that often shows species and sample position, representing maxima of species relative abundances (or probability of occurrences) and similarity in the assemblage composition of the samples, respectively. Moreover, CCA allows testing for statistical significance of the relationships between assemblage composition and independent variables. Thus, we used a randomization test to assess the statistical significance of temporal trends in CCA. Since the data formed a time series, we restricted the randomization scheme to cyclic shifts along the time sequence [33].

Finally, clinical samples were matched with samples taken in parallel from wastewaters (in 1963–2019) [23] and the relationships between the clinical and water samples were evaluated using generalized linear models (GLMs) [34]. Again, the models were fitted using binomial error distribution and logit link functions if needed, and ARMA correlation structures were also used. Since the data showed considerable overdispersion, standard errors of the GLM coefficients were adjusted by dispersion parameters. We performed these analyses for all viruses, including PV, CVB, and E, since those groups were sufficiently frequent in both clinical and wastewater samples.

The analyses were performed in R (R Development Core Team) [35] using the following libraries for creating graphs: ggplot2 [36], mgcv [30], MASS [37], and vegan [38].

## 3. Results

From 1958 to 2020 (63 years), 108,579 clinical samples were examined across the whole territory of SR. A total of 5984 EVs were isolated (5.51% of positive samples). Of those, 843 (14.09%) were PVs and 4475 (74.78%) were NPEVs. In total, 2371 of the NPEV isolates were identified only as NPEVs, while 2104 of the NPEVs were identified up to serotype level. Of the total number of samples positive for EV isolates (5984), 666 (11.13%) samples did not have serotyping data. CVBs and Es have been the prevalent NPEV viruses isolated from clinical materials in SR for a long time [15]. The GAMs revealed significant temporal trends in the probability of occurrence of all investigated virus groups. The proportion of PVs decreased steeply in the early 60s and remained low across the whole period, except in the mid-70s when a single peak occurred (estimated degrees of freedom (edf) = 8.9, F = 1627, *p* < 0.0001) (Figure 1). In contrast, the proportion of E peaked relatively recently in 2008 (edf = 8.9, F = 595.5, *p* < 0.0001). The remaining groups showed less obvious but significant patterns (CVA: edf = 8.7, F = 66.7, *p* < 0.0001; CVB: edf = 7.2, F = 130.8, *p* < 0.0001).

The trends detected at the group level were validated by the strain-level analysis (Figure 2). The most pronounced trend was the recent peak in the occurrence probability of E30 in 2008 (edf = 4.0, F = 158.5, *p* < 0.0001) and increasing occurrence probability of both E6 (edf = 3.7, F = 53.7, *p* < 0.0001) and CVB5 (edf = 3.6, F = 154.4, *p* < 0.0001). The composition of the EV records changed significantly over time (pseudo-F = 4.2, *p* = 0.0163).

The CCA showed that the composition of EVs assemblages changed significantly over time (pseudo-F = 4.2, *p* = 0.0163). The first ordination axis accounted for 6.6% of the variability in the composition data and represents the long-term shift from assemblages typical for PVs, CVB1, and CVB3 in the second half of the 20th century (left part of the ordination space), to assemblages with a higher frequency of CVA10, CVA16, E3, and E25 appearing more recently (right part of the ordination plot) (Figure 3). The second axis accounted for 7.1% of the variability and represents short-term shifts in the composition of virus assemblages. The composition of EV strains was more homogeneous in recent decades than in the 20th century, as is apparent from the lower dispersion of the sample scores along the second axis during the 21st century.

We found significant positive relationships between the proportion of all EVs (standardized regression coefficient β [95% conf. limits] = 0.49 [0.33–0.64], t = 6.0, *p* < 0.0001), CVB (β = 0.42 [0.25–0.59], t = 4.9, *p* < 0.0001), and E (β = 0.35 [0.05–0.64], t = 2.4, *p* = 0.0218) in wastewater and clinical samples (Figure 4). However, no significant relationship was found in PVs (β = 0.33 [−0.05–0.72], t = 1.7, *p* = 0.0974).

## 4. Discussion

The last wild-type PV was isolated in the SR in 1960 (WPV1 in 1960, WPV2 in 1959 WPV3 in 1959, in wastewater and in clinical samples) [15,23]. Records of EV surveillance in clinical specimens date back to the beginning of PV vaccination, which was closely followed by the launching of EV monitoring in the environment, especially in wastewater [15,23]. Earlier efforts were mainly focused on the detection of PVs. Within a few years after the vaccination program was introduced, there was a sharp decline in the incidence of the paralytic form as well as other forms of polio in CSR [13,14].

Our analyses showed that both the numbers and the serotype spectra of EVs isolated from clinical samples varied significantly over time.

Our study has a number of limitations. Historical data on isolated viral strains as well as on the methods used were obtained from archival data of laboratories and from their annual reports. The recording may have been incomplete in all laboratories in the early years especially. The data on isolates have been guaranteed to be complete since 1999, when the methods of investigation and the cell cultures used for virus isolation in the individual laboratories in the SR were harmonized in accordance with the WHO manual [26]. We did not find literature on observation and evaluation of EVs in clinical samples or in wastewater over such a long period; comparisons are therefore only partial. At the same time, the viral strains evaluated by us were obtained only via the classical cell culture method; for this reason, we could have missed some virus strains that are more difficult to culture on cell culture, e.g., CVA, EV-D68, or EV-A71. Another limitation is that the only inclusion or exclusion criteria were as defined by the EV and PV surveillance guidelines [10,26,28]. For the present study, we used all clinical cases treated through the state healthcare system, which were investigated and reported by the enterovirus surveillance system. Clinical samples were also not analyzed in terms of patient age, diagnosis, or locality, as these data were only available in a small proportion of all samples.

Incidence of PVs was highest in the early years of follow-up and declined sharply in the early 1960s. Since the early 1980s, the incidence of presence of PV Sabin-like (SL) in clinical samples has been very low, and, after a change in vaccination strategy (from OPV to IPV) in 2005, the incidence has been sporadic (Figure 1). The last isolation of PV1-SL from a clinical sample in Slovakia was in 2019.

During the study period, Es were the most abundant NPEVs in clinical samples. Their occurrence was cyclical over the studied time period, with peaks of occurrence around 1973, in the second half of the 1980s, and the highest rise at the end of the first decade after 2000. The reported increased incidence of EV-associated neurological disease in 2008, particularly in East of the SR, where E4 and E30 were the dominant serotypes, contributed significantly to this rise [39]. In 2016, an increased incidence of E6-caused disease was again noted. An increase in EV prevalence occurred approximately every 15 years, as confirmed by our analyses of E prevalence in wastewater [23]. The cyclicity of occurrence of individual E serotypes shows longer periods of time—20 years or more (Figure 2). The cyclic occurrence of Es has also been described in other studies [22]. Overall, the findings of EVs in clinical samples in the SR had an increasing trend (Figure 1). This trend was particularly pronounced for the most frequently occurring Es, which are E6 and E30, especially after 2000 (Figure 2).

The findings of CVs in clinical samples in the SR also showed a statistically significant increasing trend over time, although not as pronounced as for Es (Figure 1). The CVBs were somewhat strongly represented as compared to CVA. There was also a noticeable cyclical pattern of occurrence over time, but again less pronounced than for Es. The most abundant serotype over the entire study period was CVB5, which became significantly more prevalent after 2000, and its prevalence in clinical samples showed an upward trend (Figure 2). The significant and steadily increasing abundance of CVB5 was also consistent with findings in wastewater in the SR [23,40]. Among the CVA viruses, CVA9 and CVA16 were the dominant ones.

The highest detection rates of subtypes of NPEVs were regular, during late summer and early autumn, as reported in other studies [22]. An interesting finding is that the heterogeneity of isolated EV strains decreased with time, as shown in Figure 3. The same conclusions were reached by analyzing isolates from wastewater in Slovakia [23].

The most frequently occurring subtypes of non-polio enteroviruses in clinical samples in Slovakia during the whole period under study were CVB5, E6, and E30, which belong to *EV B* (Figure 2).

Comparison of the prevalence of EVs in clinical samples (years 1958 to 2020) to their prevalence to the wastewaters studied in parallel for 1963 to 2019 [23] showed that significant correlations were found for CVB and Es, as well as for the whole group of EVs (Figure 4). On the other hand, the occurrence of PVs in samples from clinical materials and in wastewater did not show a significant correlation. This is explained by the fact that during vaccination with the live OPV vaccine, PV-SLs were shed into wastewater after vaccination of children, as a result of which they were also regularly isolated. However, they did not cause disease; therefore, in the clinical materials (from patients of different ages, with different diseases and clinical symptoms), these PV-SLs were found only sporadically and were the predominant etiological agent of the disease in question.

Our findings of EVs in clinical samples can be compared with those from other European countries in some aspects, for example in the study on the circulation of NPEVs in 24 countries of the European Union (EU) [22] conducted between 2015 and 2017. In Germany, strains E30, E6, and EV-71 were reported to be the most frequently occurring strains from 2006 to 2019 [41]. In Kazakhstan in 2017 and 2018, CVB dominated in AFP samples in children younger than 15 years [42]. In Moldova, E30, E11, E6, and CVB1-6 viruses dominated clinical samples from 2002 to mid-2019 [43]. In Spain, during the earlier period of 1988–1997, Es (90% of NPEVs) were predominant, mainly E30, E9, E6, and E4, and among the CVs, CVB5 was predominant [44].

The published study from USA on EV surveillance during the 1970–2005 period (35 years) reported from the National Enterovirus Surveillance System (NESS) show that the most frequently occurring serotypes were E9, E11, E30, E6, and CVB5. The authors of the study report that the predominant serotypes and the order of the individual EVs have varied over time [45].

## 5. Conclusions

In conclusion, in the SR, mainly *EV B*, represented by the CVBs and Es, with frequent changes in the composition of the predominant serotypes, were observed over the studied period.

As the world moves towards complete eradication of wild PVs in the next few years, other NPEVs will naturally prevail. It is therefore still important to maintain a surveillance system, not only in terms of the GPEI gold standard of screening for patients with aseptic meningitis and AFP (especially in children under 15 years of age), but also through complementary programs such as EV surveillance and environmental surveillance for PVs and NPEVs in wastewater [46]. The SR has a long tradition in this respect. Introduction of new molecular biological and genetic methods for the detection and identification of EVs in public health laboratories in the SR needs to be continuously improved.

## Figures and Tables

**Figure 1 viruses-14-01957-f001:**
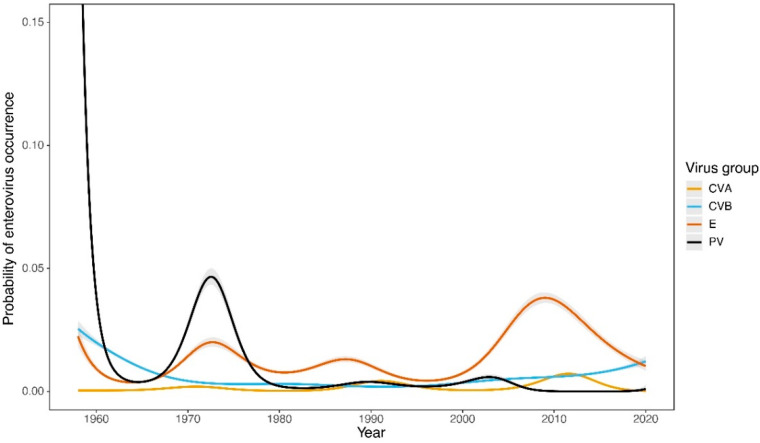
Temporal trends in probability of enterovirus occurrence in the Slovak Republic from 1958 to 2020. GAM-based estimates (lines) are displayed along with 95% confidence intervals.

**Figure 2 viruses-14-01957-f002:**
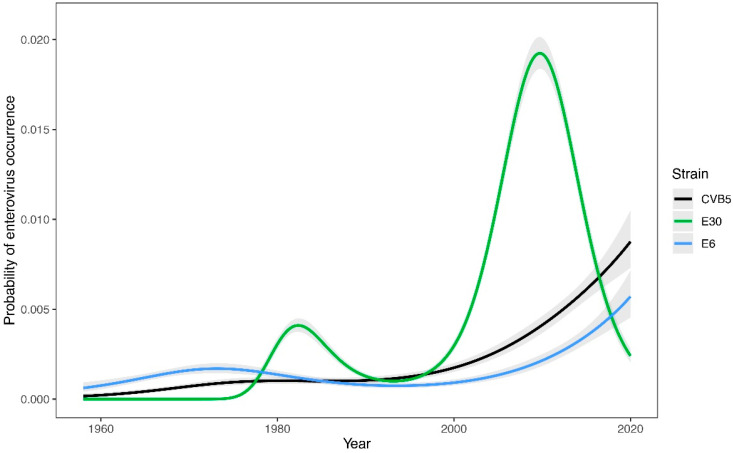
Temporal trends in probability of the three major enterovirus strains’ (coxsackieviruses (CV) and echoviruses (E)) occurrences in the Slovak Republic from 1958 to 2020. GAM-based estimates (lines) are displayed along with 95% confidence intervals.

**Figure 3 viruses-14-01957-f003:**
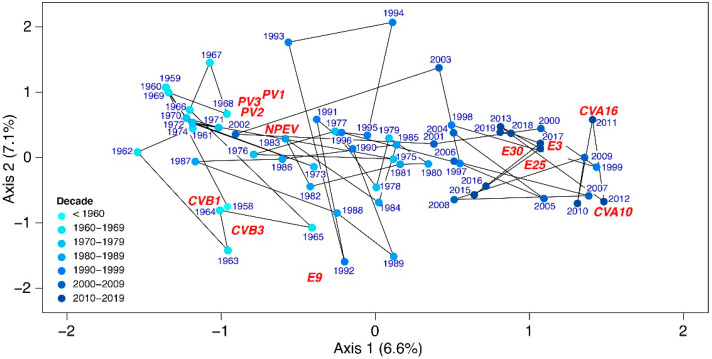
Canonical correspondence analysis (CCA) ordination plot showing temporal changes in the composition of enterovirus findings in the Slovak Republic from 1958 to 2020. Only the most frequent strains (top 50%) with the best fit (top 50%) to the temporal trend (first axis) are displayed. Ordination scores of these strains are shown in red letters (the position of the label matches the score position) and represent the space where the maxima of relative abundances are expected. Distances between sample scores (dots) represent the similarity of virus assemblage (polioviruses (PVs), coxsackieviruses (CV), and echoviruses (E)) composition among samples. The ordination plot is scaled symmetrically; variation explained by the ordination axes is displayed in parentheses. Note that the sample scores are distinguished by color in decadal steps and the axis values are standardized to zero-weighted means and unit-weighted variances.

**Figure 4 viruses-14-01957-f004:**
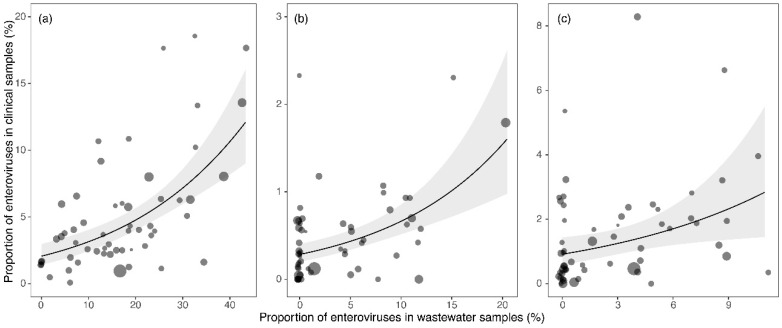
Relationships between the proportion of all enteroviruses (**a**), coxsackieviruses B (CVB) (**b**), and Echoviruses (E) (**c**) in wastewater versus clinical samples. GLM-based estimates (lines) are displayed along with 95% confidence intervals (gray bands) and observed values (points). Point size is proportional to the number of samples taken in the given year. Note that the position of samples is slightly jittered to avoid overlap. The *x* axis shows the proportion of enteroviruses in the wastewater sample in percentages, and the *y* axis show the proportion of enteroviruses in clinical samples.

## Data Availability

At the Authority of the Public Health and Regional Authorities of Public Health of the Slovak Republic.

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
