# Peer review of "Surveillance Program of Clinical Samples for Polio and Non-Polio Enteroviruses in the Slovak Republic during the 1958–2020 Period"

_viruses, 2022, doi:10.3390/v14091957_

Round 1
Reviewer 1 Report
This manuscript is well written to interpreter the regional surveillance data of enterovirus. However, there is limited description of INDIVIDUAL virus strains which many virologists care most. Recommend to take better use of those records.
Several other suggestions:
1. Line 79, CSR, only abbreviation?
2. Line 73-74, and line 78-79, redundancy or not?
3. Figure of temporal trends: please make it more clear whether the y axis value represent percent (0-100%) or absolute value (0-1)
4. Figure 1: color of CVB, E, PV too close.
5. Figure 2: highly recommend to include more viruses, especially those viruses widely known, such as CVB1-6, CVA21. Would be more comprehensive than figure 3 to most readers. These data would also provide reference to virologists that are interested in those viruses more or less prevalent.
6. Figure 3: Not friendly to readers not familiar with CCA. Rrecommend to include more details of the CCA analysis in the line 175 to 178. Do the markers PV2, PV3, PV1, do the three have the same point in the figure? Points and markers may need to be more organized.
Reviewer 2 Report
The manuscript “Surveillance program of clinical samples for polio and non-polio enteroviruses in the Slovak Republic during the period 1958-2020” describes historical data available for the surveillance of the EVs. The manuscript contains very interesting data set and attempts to analyze it by interesting methods but seems to be prepared in a hurry and lacks the maturity of a well-balanced manuscript.
The sentence in the introduction “They are excreted in the stool, over a period of days to weeks after the acute phase of infection.” refers to the length of time for secretion of the enteroviruses that have been studied in https://doi.org/10.1371/journal.pone.0201959 that can be a good reference.
The prevalence of neutralizing antibodies to poliovirus serotypes were not always equally high, but in your study the mean positivity rates were all more than 94%. Please elaborate on this issue, possibly also add information on vaccination schedules used in national vaccination program.
The introduction almost exclusively talks about the poliovirus other than general introduction on EVs in general. However, the idea is surveillance of polio and non-polio viruses. Could you please develop the introduction also to reflect this non-polio surveillance system and its importance?
The method section is missing the explanation of the population of the study and the inclusion-exclusion criteria. Also, the identification of the serotypes of the viruses were based on virus isolation and non-PCR followed by sequencing methods. This can be acceptable, but you may miss the chance to rule out the possibility of false negatives due to technical shortcomings during the development of the modern techniques. The data validity based on the assessment of each individual laboratory is a lo missing from the manuscript.
Additionally, the cell culture can favor some groups of EVs that propagate easier than some other groups such as some CVA members. This can introduce some bias in the whole result section. The success rate for virus isolation (around 5%) also reflects the fact that the isolation would underestimate the prevalence of the serotypes reported here.
The data supports the steep decline in the occurrence of EVs in early 60s. Do you have evidence that the number of samplings has not been decreased to affect the detection of the EVs in general? How about the effect of different techniques used in different time windows and their sensitivity having impact on the whole report?
You have also shown a decrease in the detection of the PVs in general after the vaccination program got into place. How about the vaccine strain circulating in the community? Do you have evidence that the viruses were not circulating?
How do you explain the proportion of the whole EV and CVB in clinical samples being less than water samples, but not similar magnitude seen in Echoviruses shown in Figure 4?
I did not get the idea behind the figure 3. Could you please be more specific about this analysis?
Discussion section is a bit lengthy and contains information that were not presented in the results section.
In general the manuscript seems to be short of design and the study questions or aims are not clearly stated or the ones meant to be met are not.
Round 2
Reviewer 1 Report
Thanks for the update! Very much appreciated. Just one minor suggestion:
Figure 3. is it possible to mark every point in the figure? If that is too crowded, how about in a supplemental figure with a format of SVG?
Reviewer 2 Report
The manuscript ” Surveillance program of clinical samples for polio and non-polio enteroviruses in the Slovak Republic during the period 1958-2020” has significantly improved and the aims and methods are clearer now.
There are still some issues especially regarding the use of Canonical correspondence analysis and related interpretations but in general this is closer to the final round of discussions now.
I shall thank the authors to consider the comments and made changes to the manuscript. It reads better, however, a moderate language check and editing is still beneficial.
Reading the manuscript for the second time it occurred that the surveillance was on symptomatic patients and yet many of the EV infections are asymptomatic. Therefore, it is good to acknowledge this issue to avoid misunderstanding.
In the line 143-144: was the use of the primary cells “human amniotic cells primary cultures from placenta) and human embryonic cells” approved by any ethical committee? Please specify this in the relevant section.
In line 144-145: Please specify if this cell line is known “human embryonic lung cells from Institute of Serum and Vaccines, Prague, the CZ have been used”.
In the line 149-150 you have mentioned that “The typing of isolates was carried out at the Institute of Epidemiology and Microbiology (IEM) Prague, CZ”. Since you have mentioned in your response that part of the molecular identification (PCR) was carried out in a lab affiliated with WHO in Finland, please add this info to the M&M section.
In the line 156: “10% calf serum”, do you mean Fetal calf serum (FCS)?
In the lines following line 160: Was media for clinical specimens contained any antimicrobial agent(s)? Please specify.
In the line 223: “MASS [37] and [38].” please modify as needed.
In the line 228: “Of the 666 (11,13%) positive samples did not have the serotyping data.” The sentence seems to be incomplete. Please modify as needed.
On the idea of the usage of “Canonical correspondence analysis (CCA)” I am not sure if the message is widely understood. For two reasons, first the graphs normally presented with the central point related to various dimensions on the plane, and second, the presence of the years and viruses on the graph does not show immediately any feeling about the time trends or composition of the assemblies. Since it is meant to be understood by a variety of readers from various disciplines, I suggest use a different representation to show the concept or explain in detail what the graph means for example what the missing info on the dots are and what the axes 2 stands for.
In the line 280: “wild poliovirus” better to be wild type poliovirus.
